# Optimal control methods for nonlinear parameter estimation in biophysical neuron models

Nirag Kadakia[1,2,3]*

**1** Department of Molecular, Cellular, and Developmental Biology, Yale University, New Haven, CT, United States of America, **2** Quantitative Biology Institute, Yale University, New Haven, CT, United States of America, **3** Swartz Foundation for Theoretical Neuroscience, Yale University, New Haven, CT, United States of America

\* nirag.kadakia@yale.edu

**Data Availability Statement:** Codes are available on Github at https://github.com/niragkadakia/oc-dspe.

**Funding:** NK was supported by a postdoctoral fellowship through the National Institute of Mental

## Abstract

Functional forms of biophysically-realistic neuron models are constrained by neurobiological and anatomical considerations, such as cell morphologies and the presence of known ion channels. Despite these constraints, neuron models still contain unknown static parameters which must be inferred from experiment. This inference task is most readily cast into the framework of state-space models, which systematically takes into account partial observability and measurement noise. Inferring only dynamical state variables such as membrane voltages is a well-studied problem, and has been approached with a wide range of techniques beginning with the well-known Kalman filter. Inferring both states and fixed parameters, on the other hand, is less straightforward. Here, we develop a method for joint parameter and state inference that combines traditional state space modeling with chaotic synchronization and optimal control. Our methods are tailored particularly to situations with considerable measurement noise, sparse observability, very nonlinear or chaotic dynamics, and highly uninformed priors. We illustrate our approach both in a canonical chaotic model and in a phenomenological neuron model, showing that many unknown parameters can be uncovered reliably and accurately from short and noisy observed time traces. Our method holds promise for estimation in larger-scale systems, given ongoing improvements in calcium reporters and genetically-encoded voltage indicators.

## Author Summary

Systems neuroscience aims to understand how individual neurons and neural networks process external stimuli into behavioral responses. Underlying this characterization are mathematical models intimately shaped by experimental observations. But neural systems are high-dimensional and contain highly nonlinear interactions, so developing accurate models remains a challenge given current experimental capabilities. In practice, this means that the dynamical equations characterizing neural activity have many unknown parameters, and these parameters must be inferred from data. This inference problem is

Health (F32MH118700), a postdoctoral fellowship through the Swartz Foundation for Theoretical Neuroscience, and a postdoctoral fellowship through the National Institute on Deafness and Other Communication Disorders (K99DC019397). The funders had no role in study design, data collection and analysis, decision to publish, or preparation of the manuscript.

**Competing interests:** The authors have declared that no competing interests exist.

nontrivial owing to model nonlinearity, system and measurement noise, and the sparsity of observations from electrode recordings. Here, we present a novel method for inferring model parameters of neural systems. Our technique combines ideas from control theory and optimization, and amounts to using data to "control" estimates toward the best fit. Our method compares well in accuracy against other state-of-the-art inference methods, both in phenomenological chaotic systems and biophysical neuron models. Our work shows that many unknown model parameters of interest can be inferred from voltage measurements, despite signaling noise, instrument noise, and low observability.

This is a *PLOS Computational Biology* Methods paper.

## Introduction

Computations in biological neural networks are shaped both by connection topologies and the response dynamics of individual neurons [1–3]. For single neurons, a versatile and biologically realistic class of computational models is built on the Hodgkin-Huxley framework—so-called conductance-based neuron models [4]. Fitting such biophysically realistic neural models to data is often cast in the framework of statistical inference [5–9], which systematically takes into account i) noise in the model dynamics and in the observations, and ii) unobservability in model states. Moreover, inference procedures produce not only an optimal estimate of model unknowns, but also the distributions of these estimates around their optima, providing a measure of estimate uncertainty. Various manifestations of statistical inference have been applied in neurobiological, behavioral, and neuromorphic settings [6, 7, 9–21].

In many settings, the emphasis of statistical inference has been on tracking *dynamical* variables, such as the membrane voltage, over time. Algorithms like the Kalman filter [5, 22] or its many variants [5, 5, 23, 24] solve this state estimation problem recursively, by updating the optimal estimate sequentially with each subsequent observation. This is computationally cheap and memory-efficient, requiring only the estimate at the most recent timestep. But extracting time-varying quantities is only one aspect of the inference procedure, and is not usually the direct quantity of interest. Rather, it is knowledge of the fixed *parameters*, such as time constants, channel conductances, baseline ionic concentrations, and synaptic strengths, that is required to generate predictions to novel stimuli, informing our understanding of brain function. Recursive filtering methods are not directly applicable when neuron model parameters are unknown, since the model dynamics, which together with the data determine subsequent estimates, are not fully specified.

One approach for parameter estimation in filtering is to "promote" parameters to dynamical states, but with trivial dynamics—this keeps parameters approximately constant, while allowing enough stochasticity to improve estimates [5, 25]. More sophisticated methods combine filtering and parameter inference into an expectation-maximization (EM) algorithm that updates state and parameter estimates in alternating fashion [5, 26]. However, such techniques have been limited to the estimation of linear parameters such as ion conductances, rather than nonlinear parameters such as those governing ion channel kinetics. This is due to the fact that EM algorithms converge to local minima; with nonlinearities and noise, cost manifolds are highly nonconvex, precluding reliable estimation.

An alternative to filtering approaches is the optimization of a posterior distribution over, jointly, the states at all timepoints and the unknown parameters. Optimization-based methods

have been used to effectively estimate linear parameters in neural models [6, 26–30], as well as to uncover nonlinear parameters in nonlinear and even chaotic models [7, 17–20, 31–37]. Here, we build on some of these approaches to present a new method for joint state and parameter estimation in biophysical neural models. Our emphasis is on strongly nonlinear (including chaotic) models, sparse observations, unknown nonlinear parameters, and extreme nonconvexity in the state and parameter manifold. We combine two ideas which have recently been investigated in nonlinear systems estimation: chaotic synchronization [38–40], and homotopy continuation [7, 33, 41, 42]. Our approach is predicated on a previous observation that, by coupling measured data directly into the system dynamics, cost manifolds over the unknown states and parameters become highly regularized [37, 40]. We extend this notion into an optimal control framework, treating the coupling strength as a control parameter which we optimize using the Pontryagin minimum principle [43]. What results is a new dynamical system that represents the evolution of the optimal state estimate in time. We present an algorithm to simultaneously find the integral curves of these "estimation dynamics" and determine the unknown parameters from noisy data. We illustrate our algorithm using synthetic data from both a sparsely observed canonical choatic attractor and a biophysical HH-type model, comparing to the prior techniques [33, 40].

We find that parameter estimates in nonlinear and chaotic systems are significantly more accurate than in previous estimation methods, particularly for very sparse observations. Our technique is able to accurately estimate nonlinear model parameters from short time traces, improving on prior efforts to uncover linear parameters [26]. Moreover, estimates are considerably robust to both measurement noise and model noise. The latter manifests as noisy current inputs, affecting the timing of action potentials, and has not been investigated extensively in other optimization-based approaches. In sum, our method finds significant improvements in prediction accuracy over other optimization-based estimation methods, while exhibiting some tradeoffs in higher computational cost due to additional constraints.

## Results

### Nudging synchronization for estimating dynamical states

We first consider the problem of estimating the hidden dynamical states of a system with known dynamics from noisy observations, focusing on a simple technique which will form the basis for our proposed approach. The system evolves either deterministically via the ODE dynamics

$$\dot{\mathbf{x}}(t) = \mathbf{f}(\mathbf{x}(t), \mathbf{\Theta}) \tag{1}$$

or stochastically under the Langevin dynamics

$$\dot{\mathbf{x}}(t) = \mathbf{f}(\mathbf{x}(t), \mathbf{\Theta}) + \eta(t), \tag{2}$$

where $\eta(t)$ is uncorrelated Gaussian noise, and for now we assume that all model parameters $\mathbf{\Theta}$ are known. The system is sparsely observed, so the dimension $L$ of the observation vector $\mathbf{y}(t)$ is generally less than that of the $D$-dimensional state space $\mathbf{x}(t) = [x_1(t), x_2(t), ..., x_D(t)]$, e.g. $L < D$. For notational simplicity, throughout we represent $\mathbf{y}(t)$ as a $D$-dimensional vector with non-zero elements only in its $L$ observed elements. An unsophisticated but straightforward and computationally attractive way to infer the evolution of both the observed and unobserved variables is by dynamically coupling the system to the measurements, effectively controlling the system dynamics with data [38, 40, 45–47]. To do this, we redefine the dynamics as:

$$\dot{\mathbf{x}}(t) = \mathbf{f}(\mathbf{x}(t), \mathbf{\Theta}) + \mathbf{U} \cdot (\mathbf{y}(t) - \mathbf{H}\mathbf{x}(t)) \equiv \mathbf{f}_{\text{cont}}(\mathbf{x}(t), \mathbf{y}(t), \mathbf{\Theta}) \tag{3}$$

where $\mathbf{U}$ is a constant, diagonal matrix with nonzero elements $u_{ll}$ for observed states $l$, and $\mathbf{H}$

projects the state space $x$ onto the observed subspace $\mathbf{y}$—i.e. both $\mathbf{U}$ and $\mathbf{H}$ are $D$x$D$ but have rank $L$. The initial states $\mathbf{x}(t_0)$, can be initialized randomly, and the dynamics Eq 3 are simply integrated forward [45]. Information passes from the data to the observed $L$ states through the linear control term $\mathbf{U}(\mathbf{y} - \mathbf{Hx})$, then from observed to the $D$—$L$ unobserved states through the coupled dynamical equations (Fig 1A). Information therefore passes from data to *all* model states, observed or not. The strength of the nonzero control terms $u_{ll}$ should be set to a value large enough to synchronize the system to the data, but not so large that the noise in the observations is magnified [46]. In the geophysics literature, this technique is known as "nudging" [45, 46]. Those familiar with linear filtering or control theory will recognize that the control term, a linear scaling of the error between data and estimate, is analogous to the innovation term in the Kalman filter [5, 22]. However, the Kalman gain evolves in time and is chosen to minimize the residual estimation error at each time step, while $\mathbf{U}$ is constant and prescribed upfront. Though the Kalman prescription is more principled than the choice of $\mathbf{U}$, there is a tradeoff in the computational costs of storing and manipulating large convariance matrices, not required here. Still, the more pressing issue is that sequential estimators such as nudging and Kalman filters are not directly suited for the estimation of static parameters. At each timepoint, sequential estimators incorporate new observations with both the running estimate from the previous timestep and the model predictions; without known parameters, the latter cannot be obtained.

We illustrate nudging synchronization in the chaotic Lorenz96 [44] system in 5 dimensions:

$$\dot{x}_d = (x_{d+1} - x_{d-2})x_{d-1} - x_d + F$$
$$d = 1, .., 5; \quad x_{d+5} = x_d \tag{4}$$

The Lorenz96 system contains a single parameter $F$, which for values $\sim 8$ render the dynamics chaotic. We assume that only $x_1$ and $x_4$ are observed, with Gaussian measurement noise, so

$$\mathbf{y} = \mathbf{Hx} + \mathbf{R}(t) \tag{5}$$

$$H_{11} = H_{44} = 1$$
$$R_1(t), R_4(t) \sim \mathcal{N}(0, \sigma^2)$$

and the remaining $H_{ij}$ are zero and $\sigma = 1$. True states $\mathbf{x}$ are generated by integrating Eq 4 numerically with a timestep of $dt = 0.01$ over $t \in [0, 5]$, from which observations are generated using Eq 5 (Fig 1B). Using these observations $\mathbf{y}$, we obtain an estimate $\hat{\mathbf{x}}$ of the true states by integrating forward the nudged dynamics, Eq 3, with $U_{11} = U_{44} \equiv u$ set to a fixed value, chosen between 0 and 100. We initialize our estimate at time $t = 0$ to the measured data $\mathbf{y}(0)$ for the observed variables $x_1$ and $x_4$, and uniformly between ±5 for the hidden variables. This chaotic system is hypersensitive to errors in the initial conditions, so without control, $u = 0$, the estimated trajectory evolves in a manner quite distinct from the true model (Fig 1C). For sufficient $u$, e.g. $u = 5.0$, the data synchronizes the estimates of both observed and unobserved states to the model, despite large initialization errors in $\hat{\mathbf{x}}(0)$ (Fig 1D). A distinct advantage of nudging is that fine-tuning $u$ is not necessary: within a substantial window of $u$, between $\sim 5$–$20$, the system closely synchronizes to the true states; however, for $u$ sufficiently large, the control term now overamplifies the observation noise, degrading the estimates (Fig 1E). The simplicity of nudging synchronization is apparent: it necessitates only the choice of the gain parameter $u$, which can be chosen rather loosely—and requires only a simple forward integration of the model dynamics.

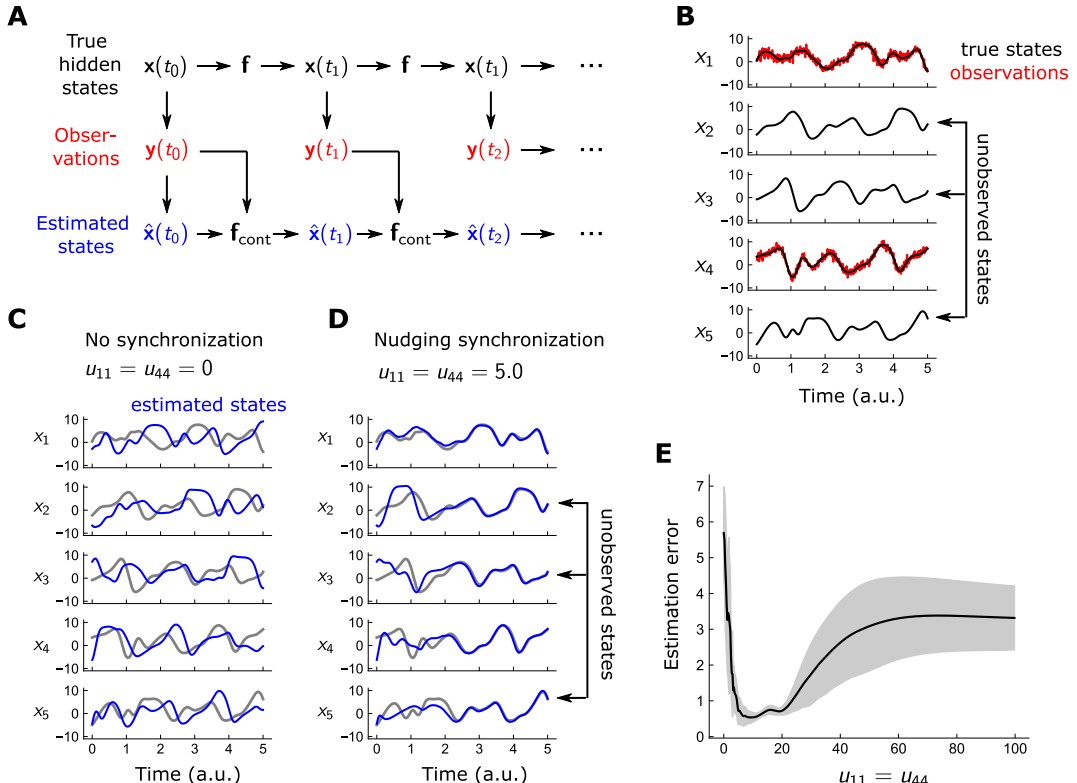

**Fig 1. Nudging synchronizes estimate to observations with lax parameter tuning.** A: Hidden states $\mathbf{x}$ evolve temporally under the controlled dynamics via $\dot{\mathbf{x}} = \mathbf{f}_{\text{cont}}(\mathbf{x})$, where $\mathbf{f}_{\text{cont}}$ is the sum of model dynamics and an external data-driven control term. The control feeds back the difference between $\mathbf{x}$ and the observations $\mathbf{y}$, for measured states; it is absent in unmeasured states. B: Ground-truth trajectories and observations Lorenz96 system in 5 dimensions; only $x_1$ and $x_4$ are observed. Measurements obtained by adding uncorrelated noise with $\sigma = 1$ at each timestep. C and D: True (black) and estimated (blue) trajectories of the Lorenz96 system, for no nudging control (C) and with $x_1$ and $x_4$ controlled with strength $U_{11} = U_{44} \equiv u = 5$ (D). The controlled system synchronizes closely to ground truth for both the observed unobserved variables. E: Mean-squared error $\frac{1}{ND} \sqrt{\sum_{d,n} (x_d(t_n) - \hat{x}_d(t_n))^2}$ of estimates as a function of control strength; near-minimal errors are achieved over a relatively wide range of $u$.

## Parameter inference in nudging synchronization

In the context of neuroscience, we are less interested in the time course of dynamical state variables than model parameters such as ion channel conductances and kinetic time constants [18, 19, 48]. One could envision a direct search over parameter space with some appropriately chosen cost function. To illustrate why a naive optimization may not be entirely straightforward, we plot the quadratic cost between the measurements and model, $C(F) = \sum_n ||\mathbf{y}(t_n) - \mathbf{H}\mathbf{x}(t_n; F)||^2$, where $\mathbf{x}(t_n; F)$ is the solution of the Lorenz96 model with forcing parameter $F$ (Fig 2A) and the initial condition $\mathbf{x}(t_0; F)$ is fixed to its true value. The global minimum of this cost surface for $F$ corresponds to the true parameter $F_{\text{true}} = 8$, but it resides in a narrow basin of attraction and is surrounded by multiple false minima (Fig 2A). This irregularity is a consequence of the highly nonlinear nature of the dynamics [37, 40], and it would be exceedingly difficult to pinpoint the global minimum with conventional optimization routines, even in this optimistic scenario in which $\mathbf{x}(t_0)$ is assumed known and the search space is effectively one-dimensional.

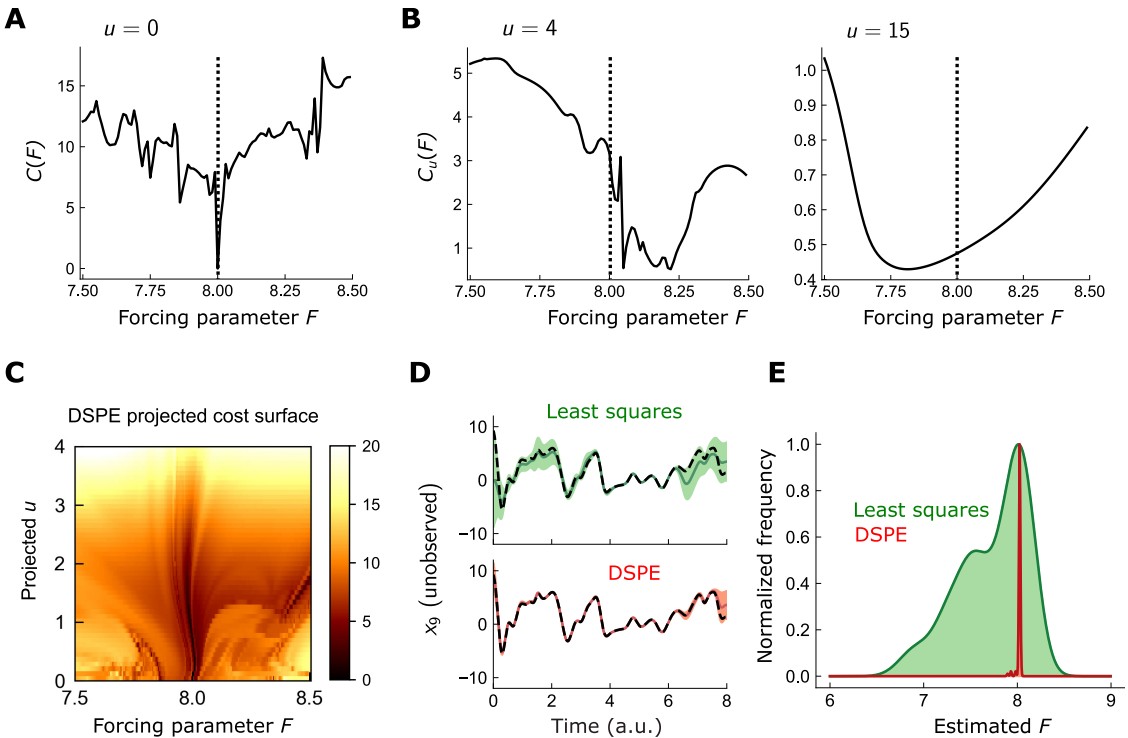

**Fig 2. Constraining least-square estimates with controlled dynamics enables robust parameter estimation in chaotic models.** A: Squared-difference between 10D Lorenz96 model trajectory **x** and measured data **y**, for various values of forcing parameter $F$, assuming that model was integrated forward from the true initial point. Only states $x_1$, $x_4$, $x_7$, $x_{10}$ were measured. The global minimum of $C(F)$ (dotted line; $F=8$), resides in an extremely narrow basin of attraction within a highly nonconvex cost surface. B: Same as A, but for controlled dynamics ($u = 4$ and $u = 15$ for middle and right plots, respectively). Higher controls provide a far smoother cost function around the global minimum. C: 2D projection of the high-dimensional DSPE cost surface, along $F$ in one dimension and the line $U_l(t_n) \equiv u$, $\forall l$, $n$ in the other. Here, $x_d(t_n)$ were fixed at the values generated by integrating the model forward from its true initial state for the given $F$. D: Estimated trajectories of $x_8$, using uncontrolled least squares (top; green) and DSPE (bottom; red). Black line: true trajectory. Shaded regions: SD of estimate across 100 random initializations of the optimization. E: Histogram of parameter estimates across 100 initializations of the optimization, for least squares (green) and DSPE (red).

On the other hand, the cost between the measurements and the *controlled* dynamics is far more regular. If we instead consider $C_u(F) = \sum_n ||\mathbf{y}(t_n) - \mathbf{H}\mathbf{x}(t_n; F, u)||^2$, where $\mathbf{x}(t_n; F, u)$ is generated from the controlled dynamics Eq 3, we find that the cost surface smooths considerably with increasing nudging strength (Fig 2B). For sufficient gain ($u = 15$), the cost surface is fully convex, exhibiting a broad minimum around $F = 7.81$. Thus, the linear control term proportional to $u$ acts to smooth the nonconvexities of the cost surface induced by the chaotic dynamics. This observation led Abarbanel et al. to propose the following constrained optimization problem [40]:

$$C(\mathbf{x}(t_n), \mathbf{\Theta}, \mathbf{U}(t_n)) = \sum_n ||\mathbf{H}\mathbf{x}(t_n) - \mathbf{y}(t_n)||^2 + ||\mathbf{U}(t_n)||^2, \qquad (6)$$

$$\text{s.t.} \quad \dot{\mathbf{x}}(t_n) - \mathbf{f}_{\text{cont}}(\mathbf{x}(t_n), \mathbf{U}(t_n), \mathbf{\Theta}) = 0, \quad \forall n \qquad (7)$$

where, in contrast to direct nudging, the control terms are now time-dependent, $\mathbf{U} \to \mathbf{U}(t_n)$. In this *dynamical state and parameter estimation* (DSPE) framework, the dynamical states $\mathbf{x}(t_n)$, control terms $\mathbf{U}(t_n)$, and model parameters $\mathbf{\Theta}$ are optimized simultaneously. With no control ($\mathbf{U}(t_n) = 0$), this optimization reduces to a naive constrained least squares matching of the

model to the observations. The insight of DSPE is that the cost function becomes smoother for larger $\mathbf{U}(t)$, localizing the estimate to the vicinity of the global minimum, while the quadratic penalty on $\mathbf{U}(t)$ reduces the controls, refining the estimate to the true minimum of the uncontrolled dynamics. Thus, one can think of $\mathbf{U}(t)$ as opening up new degrees of freedom, allowing escape from local minima during the optimization procedure. When the optimization has terminated, the optimal $\mathbf{U}(t_n)$—which are driven to small values due to the $\sim \mathbf{U}^2$ cost penalty— are discarded. In this sense, they are essentially an algorithmic device.

We illustrate the regularizing effects of the control parameters by plotting a 2D projection of the DSPE cost surface, along $F$ and along a 1D projection in the space of $\mathbf{U}(t_n)$ (Fig 2C): the surface has a broad minimum for larger control strength, and a complex, rugged surface as the control strength becomes weaker. To illustrate the performance of the DSPE algorithm, we compare it against a naive constrained least squares optimization without controlled dynamics, again using the 10D Lorenz96 system. Across 100 random initializations, both least squares and DSPE produced comparably accurate estimates of the state variables, with DSPE being somewhat more accurate near the boundaries of the time window (Fig 2D). The differences in the estimate of the forcing parameter $F$, however, were more striking. For nearly all initializations, DSPE estimated $F$ within 1% of its true value, while the distribution of parameters estimated by least squares peaked at the true value but were far more dispersed (Fig 2E).

## Optimally-controlled dynamical parameter inference

We now present our main results. We have seen that DSPE is built naturally on nudging synchronization, enforcing controlled dynamics as constraints in a global optimization over states, parameters, and time-dependent controls. Here, we suggest that the control variables, rather than acting as an algorithmic device to regularize the optimization procedure, could be instead specified in a more principled manner, by viewing DSPE as the foundation for an optimal control problem. In this framework, the optimal estimate can be derived using the Pontryagin minimum principle. Using the DSPE cost function Eq 6, expressed in continuous time,

$$C = \int_0^T dt \mathcal{L}(\mathbf{x}, \mathbf{U}) = \int_0^T dt \left[ \frac{1}{2} ||\mathbf{Hx} - \mathbf{y}||^2 + ||\mathbf{U}||^2 \right], \tag{8}$$

and the controlled dynamics Eq 3, we apply the minimum principle in the usual way (Methods for derivation) to obtain an expression for the optimal control (assuming $\mathbf{U}$ is diagonal for readability; this can be relaxed trivially):

$$U_{ll} = -p_l[\mathbf{y} - \mathbf{Hx}]_l, \quad (\text{for } x_l \text{ observed}). \tag{9}$$

This control can be used to derive the dynamics obeyed by the optimal estimate and the *conjugate momenta $p_d$* in time:

$$
\begin{aligned}
\dot{x}_d^* &= f_d(\mathbf{x}^*, \mathbf{\Theta}) - p_d^*([\mathbf{y} - \mathbf{Hx}^*]_d)^2 \\
\dot{p}_d^* &= -\frac{\partial \mathbf{f}(\mathbf{x}^*, \mathbf{\Theta})}{\partial x_d^*} \cdot \mathbf{p}^* + [\mathbf{H}^T(\mathbf{y} - \mathbf{Hx}^*)]_d (1 - (p_d^*)^2)
\end{aligned} \tag{10}
$$

where the asterisk refers to the fact that the integral curves of these equations represent locally optimal trajectories. Since this dynamical system describes the evolution of the optimal estimate, we call it the *estimation dynamics*. Note that the estimation dynamics form a boundary value problem rather than an initial value problem [49]: $x_d$ in the null-space of $\mathbf{H}$ are unobserved, so their values at $t = 0$ are unspecified. A wealth of studies have been devoted to solving

control systems of this class, using polynomial approximations—so-called collocation methods—or instead using *shooting* methods, which solves the initial value problem perturbatively until the boundary conditions are matched [50]. Our problem has the added complication that with unknown parameters, the dynamics could not be integrated forward in either case. Instead, we propose an algorithm for simultaneously estimating $\mathbf{x}$, $\mathbf{p}$, and $\boldsymbol{\Theta}$ that leverages some desirable aspects of the original cost function $\mathcal{L}(\mathbf{x}, \mathbf{U})$ in a practical implementation. We first use the optimal control condition Eq 9 to express $\mathcal{L}$, Eq 8, in $\{\mathbf{x}, \mathbf{p}\}$ space:

$$\tilde{\mathcal{L}}(\mathbf{x}, \mathbf{p}) = \sum_l \frac{1}{2} \left[ ([\mathbf{y} - \mathbf{Hx}]_l)^2 (1 + (p_l)^2) \right] \tag{11}$$

This is the original cost function to be optimized, now written in $\mathbf{x}$, $\mathbf{p}$. Since the estimation dynamics Eq 10 *fully* define the optimal trajectory, $\tilde{\mathcal{L}}$ contains no new information [49]. Nevertheless, this function must be stationary along locally optimal trajectories. Further, it is convex in the space of observable $\mathbf{x}$ and associated $\mathbf{p}$, so its minimum is global in those directions, and highly degenerate in the unmeasured directions.

We could therefore use the solutions of $\min \tilde{\mathcal{L}}$ as a *starting point* for the more complicated task of satisfying the highly nonlinear estimation dynamics. Specifically, we could begin by optimizing $\tilde{\mathcal{L}}(\mathbf{x}, \mathbf{p})$, subject to a loose enforcement of Eq 10. The resultant path and parameters would then be used as the initialization for a subsequent minimization of $\tilde{\mathcal{L}}(\mathbf{x}, \mathbf{p})$ subject to Eq 10, now enforced to a tighter degree. We call this method "optimally-controlled dynamical state and parameter estimation" (OC-DSPE), and schematize it in Algorithm 1. The idea of OC-DSPE is that the enforcement of the nonlinear dynamics breaks the convexity in a gradual manner, allowing the global minimum to be systematically tracked—in much the same spirit as the control variable directions in the DSPE cost manifold (vertical direction in Fig 2C). This iterative procedure lies in a class of nonlinear techniques called *homotopy continuation*, where the solutions of one system are tracked by iterating from a simpler system with known solutions [41]. Incorporating this flavor of homotopy continuation into nonlinear estimation was the basis of a number of studies, in particular *quasi-static variational assimilation* [51] and *variational annealing* [7, 32–34]. The latter has been shown to be quite effective in tracking highly chaotic systems, as well as in highly under-observed neuron models with many unspecified nonlinear parameters. A more recent study illustrated benefits in nonlinear system estimation by extending variational annealing to Hamiltonian manifolds [49], in much the same spirit as Algorithm 1. OC-DSPE differs from these other homotopy continuation techniques essentially in its integral equations Eq 10, which are derived from a set of *controlled* state space dynamics. In previous formulations, the dynamics are uncontrolled.

**Algorithm 1** Optimally-controlled dynamical state and parameter estimation (OC-DSPE)

```
Given measured data yₙ and annealing parameters βₘₐₓ, α, λ₀
for q = 1, ..., Q, in parallel do
    Sample x�q uniformly from presumed dynamical range
    Sample p�q uniformly from presumed range
    Sample Θq from prior distribution
end for
for β = 1, ..., βₘₐₓ do
    λ ← λ₀ αᵇ
    for q = 1, ..., Q in parallel do
```
$$g_x(\mathbf{x}^q, \mathbf{p}^q, \boldsymbol{\Theta}) := \text{discretization of} \left[ \dot{x}_d^q - f_d(\mathbf{x}^q, \boldsymbol{\Theta}) + p_d^q ([\mathbf{y} - \mathbf{Hx}^q]_d)^2 \right]$$
$$g_p(\mathbf{x}^q, \mathbf{p}^q, \boldsymbol{\Theta}) := \text{discretization of} \left[ \dot{p}_d^q + \frac{\partial \mathbf{f}(\mathbf{x}^q, \boldsymbol{\Theta})}{\partial x_d^{Tq}} \cdot \mathbf{p}^q - [\mathbf{H}^T (\mathbf{y} - \mathbf{Hx}^q)]_d (1 - (p_d^q)^2) \right]$$
$$\{\mathbf{x}^q, \mathbf{p}^q, \boldsymbol{\Theta}\} := \arg\min [\mathcal{L}(\mathbf{x}^q, \mathbf{p}^q) + \lambda g_x(\mathbf{x}^q, \mathbf{p}^q, \boldsymbol{\Theta}) + \lambda g_p(\mathbf{x}^q, \mathbf{p}^q, \boldsymbol{\Theta}^q)]$$

```
        return x^q, p^q, Θ^q
    end for
end for
```

## Numerical experiments of OC-DSPE with 10-dimensional Lorenz96 model

We first apply OC-DSPE to the Lorenz96 system in 10 dimensions with unknown forcing parameter *F*, performing 100 optimizations with different initial guesses (Methods for implementation details). We first show illustrative plots of the state estimates, assuming *M* = 4 states are observed. For the observed $\mathbf{x}_1$, the state estimate averaged over all 100 estimations is well-localized around the true trajectory (Fig 3A; top plot). The unobserved state variable is markedly less accurate (Fig 3B; top plot), but does approximate certain regions well (i.e. $t \sim 4$–6), suggesting that the chaotic instabilities are less pronounced in those regions around the attractor. Of course, since the trajectories depend on the parameters, the accuracy of the states is often correlated with that of the parameters. If we plot the state estimates for which the forcing parameter is estimated close to its true value, $|\hat{F} - F_{\text{true}}| < 0.05$, we find that the states are nearly indistinguishable from the true trajectories, beyond an initial synchronization region, $t > \sim 1$ (Fig 3A and 3B; bottom). This initial synchronization region is not surprising. The optimal control quickly nudges the estimate toward the observed data, even if the state at the beginning of the estimation window *t* = 0 is highly inaccurate. Once synchronized, the optimal control maintains the estimate near the true trajectory at minimal control cost $\mathbf{U}^2$.

Next, we focus on the accuracy of parameter estimates and compare i) OC-DSPE, ii) DSPE, and iii) constrained least squares (see Methods for details of the implementation of each method). We repeated the 100 estimations for *Z* = 100 distinct estimation windows around the Lorenz attractor to get a better representation of the estimation in regions where the chaotic instabilities are both strong or weak. We first plot the distribution of estimated parameters when *M* = 5 states ($x_d$; *d* odd) are observed. For *M* = 5, the estimated parameters are highly localized around the true value for all techniques (Fig 3C). The advantage of OC-DSPE becomes more apparent with fewer measured states. In particular, with *M* = 2 observed states, the estimates for all parameters are substantially more dispersed (Fig 3D). Only OC-DSPE, however, is centered near the true value; DSPE in particular is highly dispersed giving erroneous parameter estimates as low as 0. This illustrates that OC-DSPE, while computationally more demanding (the state space is doubled and the equations require more derivatives), can produce substantially more accurate parameter estimates in very sparsely observed chaotic systems.

To further quantify the robustness of the estimation procedure, we calculated the estimated parameter as a function of the error between observations and estimates,

$E = \sqrt{\sum_n ||\mathbf{H}\hat{\mathbf{x}}(t_n) - \mathbf{y}(t_n)||^2}$ (Fig 3E). Here, we chose, for each of the *Z* estimation windows, the optimal parameter estimate among the 100 initializations. This gives a "best-case" estimate for each dataset, and quantifies how this optimal parameter estimate depends on the errors in the dynamical states. Though all 3 algorithms produce accurate $\hat{F}$ when the estimation error is minimal, OC-DSPE is far more tightly centered on the true parameter value even as the state error increases. This indicates that for OC-DSPE i) the optimal parameter estimate could be found more reliably with less initializations and ii) the optimal estimate can be identified with more reliably in practice, where the only error metric available is *E*.

Finally, we consider the role of the auxiliary momenta variables *p* in OC-DSPE, a feature absent in direct optimization schemes. In the control-theoretic sense, *p* represent the marginal cost of violating the constraints given by the system dynamics (Eq 7). Regions in which *p* are appreciable correspond to regions in which the state estimate could easily deviate from the

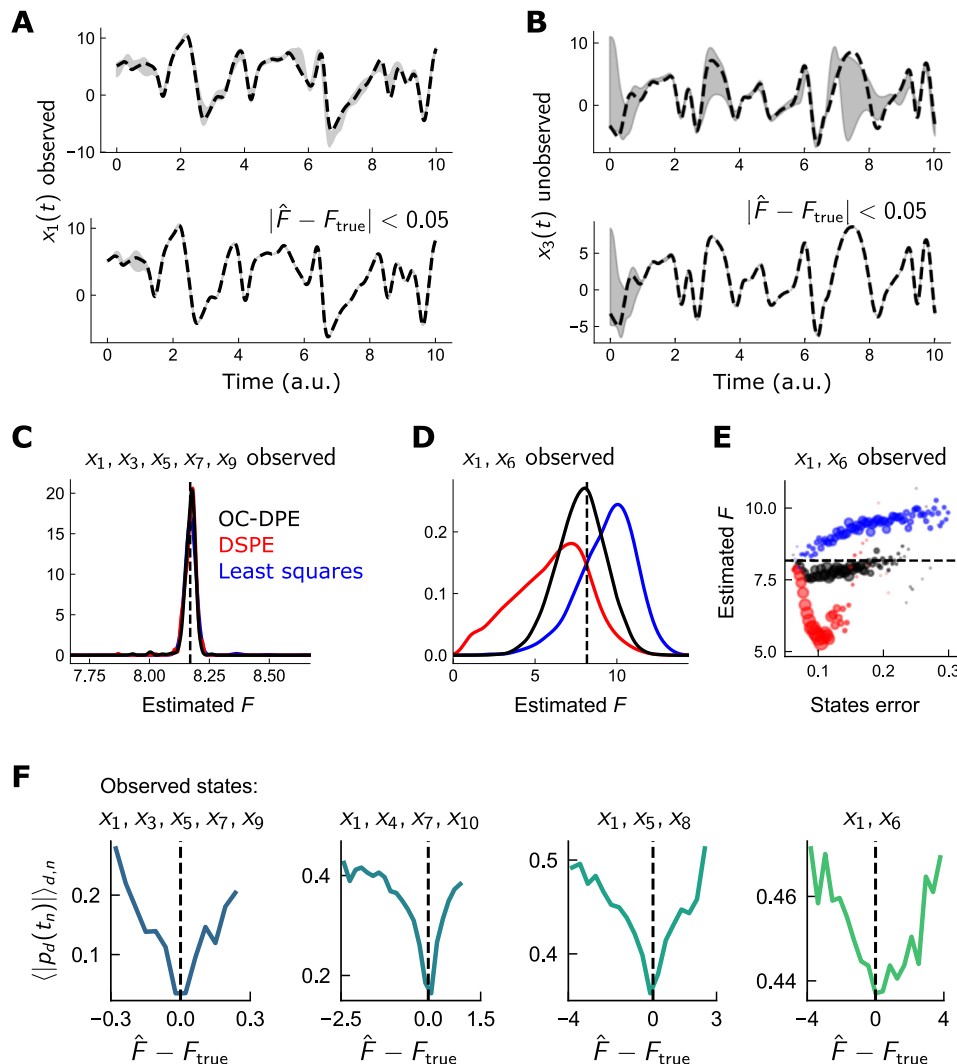

**Fig 3. OC-DSPE exhibits enhanced parameter estimation accuracy in sparsely-observed chaotic models.**
Illustration of parameter estimation in chaotic Lorenz96 system using optimally-controlled dynamical state and parameter estimation (OC-DSPE). A: Illustrative estimate of observed variable $x_1(t)$ when 4 states are observed, for 100 estimations randomly initialized (top), and for only those estimations in which the parameter was estimated to high accuracy (bottom; $|\hat{F} - F| < 0.05$). Grey: standard deviation of all estimates; Dotted: true state. B: Same as A, for the unobserved variable $x_3(t)$. C: Histogram of parameter estimates over 100 initializations, for 100 datasets initialized throughout the 10D Lorenz96 chaotic attractor, for 5 observed variables and using 3 different estimation routines (least squares, the original DSPE algorithm, and OC-DSPE). Parameter estimates are tightly clustered around the true value of 8.17. D: Same as C, for only 2 observed variables. Estimated $\hat{F}$ is more widely dispersed, but centered on the true value only for OC-DSPE. E: Estimated parameter $\hat{F}$ versus error in observed states $x$ averaged over space and time. For low state error, estimated parameters are close to the true value for all 3 estimation routines. Tor larger state errors, proximity to the true value occurs only in OC-DSPE, indicating a larger degree of robustness in estimation accuracy. F: Dependence of $|p_d(t_n)|$, averaged over dimension $i$ and time $n$, on error in parameter estimate, $\hat{F} - F$, for 5 (leftmost plot), 4, 3, and 2 (rightmost plot) observed states. In all cases, the minimum average $|p|$ occurs at very close to the true $F$, indicating that the values of the conjugate momenta can act as a proxy for identifying the optimal parameter estimate.

controlled dynamics, suggesting a lack of estimation accuracy (Fig 3E). Do deviations in $p$ also have some bearing on the reliability in the parameter estimate? In other words, is there a correlation between some statistic of $p$ and the estimation errors in $F$? Indeed, plotting the magnitude of $p$ averaged over time and dimension as a function of $\hat{F} - F_{\text{true}}$ shows a prominent dip

at zero parameter estimate error (Fig 3F). Though this dip is more prominent for larger observability (leftmost plot in Fig 3F)), it is still present even for very sparse observability (rightmost plot in Fig 3F)). This indicates that the statistics of such artificial "conjugate" variables can provide a further consistency check on the accuracy of the estimation.

## Numerical experiments with the Morris-Lecar model

Our primary system of interest is a biophysically realistic neuron modeled on the Hodgkin-Huxley framework. We use the Morris Lecar (ML) system [1], a reduced 2-state variable spiking neuron model in which the gating variable $w(t)$ drives changes in the neuron membrane voltage $V(t)$, and which has 12 static parameters (Methods). We first use OC-DSPE to estimate only the linear conductances $g_i$, which dictate the regimes of neuron excitability; we hold the other parameters fixed at their correct values. Observations $\mathbf{y}(t) = \{V_{obs}(t)\}$ are generated by adding uncorrelated Gaussian noise $\sim \mathcal{N}(0, \sigma)$ to the ground truth voltage $V_{\text{true}}(t)$, where we investigate $\sigma = 2$ mV and $\sigma = 10$ mV. The stimulating current is held at a constant value (Fig 4A). To illustrate the robustness of the algorithm to parameter uncertainty, we let the optimization bounds on the unknown conductance parameters span 2 orders of magnitude, from 0 to 200. To cross-validate our predictions, we forward integrate the dynamics from the final time estimate state $\hat{V}(T), \hat{w}(T)$ using the estimated parameters $\hat{g}_i$ but a pseudo-noisy current spanning a range of input currents (Fig 4A). We use $Q = 25$ initializations in Alg. 1.

For small measurement noise, $\sigma = 2$, most of the $Q$ runs give highly accurate estimates of both the states $V(t)$ (Fig 4B and 4C) and parameters $g_i$ (Fig 4D). For the poorest fit, $g_i$ is sufficiently inaccurate such that spike events are occasionally missed. For higher measurement noise, $\sigma = 10$ mV, about 10% of the dynamic range, many predictions have degraded as expected, producing shifted or missed spike times (Fig 4E). Still, the optimal prediction and estimated parameters are highly accurate among these 25 initializations (Fig 4E).

To get a sense of how OC-DSPE compares to related estimation methods, we repeated the estimations using the original DSPE method and constrained least squares (as in Fig 3). For small measurement noise ($\sigma = 2$ mV), the optimal predictions among $Q = 25$ runs were equally as accurate as OC-DSPE (Fig 4G), although the likelihood of finding the optimal fit was far less than the near 100% likelihood of OC-DSPE (Fig 4F). The accuracy degraded precipitously with measurement noise, and even the best parameter estimates for these two methods were poor (Fig 4F), producing considerably misshappen spike waveforms (using DSPE) or wildly inaccurate estimations altogether (using least squares) (Fig 4G). This indicated that even in this relatively tractable problem of estimating linear parameters from noisy data, OC-DSPE exhibits a robustness and accuracy not evident in related estimation methods.

As a more realistic numerical experiment, we next estimated all 11 model parameters governing ion channel kinetics (we omit the capacitance $C$, which is typically ascertained from neuron size). All parameters are again bounded liberally, over 2 orders of magnitude. For low measurement noise, the model parameters, including those entering the model equations nonlinearly, are estimated to high precision—this is borne out in the accuracy of the forward predictions (Fig 5A and 5B). Still, as before, spikes can be missed and/or slightly shifted (Fig 5C)).

As noted in a number of prior studies [18, 19, 31], the accuracy of the estimation is inherently limited by the richness of the driving current. If, for example, the stimulating current were 0 pA—below the spiking threshold—the estimated conductances would be degenerate: the fixed point of $\dot{V}$ would be unchanged by appropriate rescalings of $g_i$. Difficulties in estimating the kinetic parameters such as $\beta_i, \gamma_i$ would also arise since there are no observations when the neuron is spiking. In the case of a single step, as in Figs 4A and 5A, the volume of phase space in $V(t), w(t)$ occupied by the system is vanishingly small, covering only the closed curve

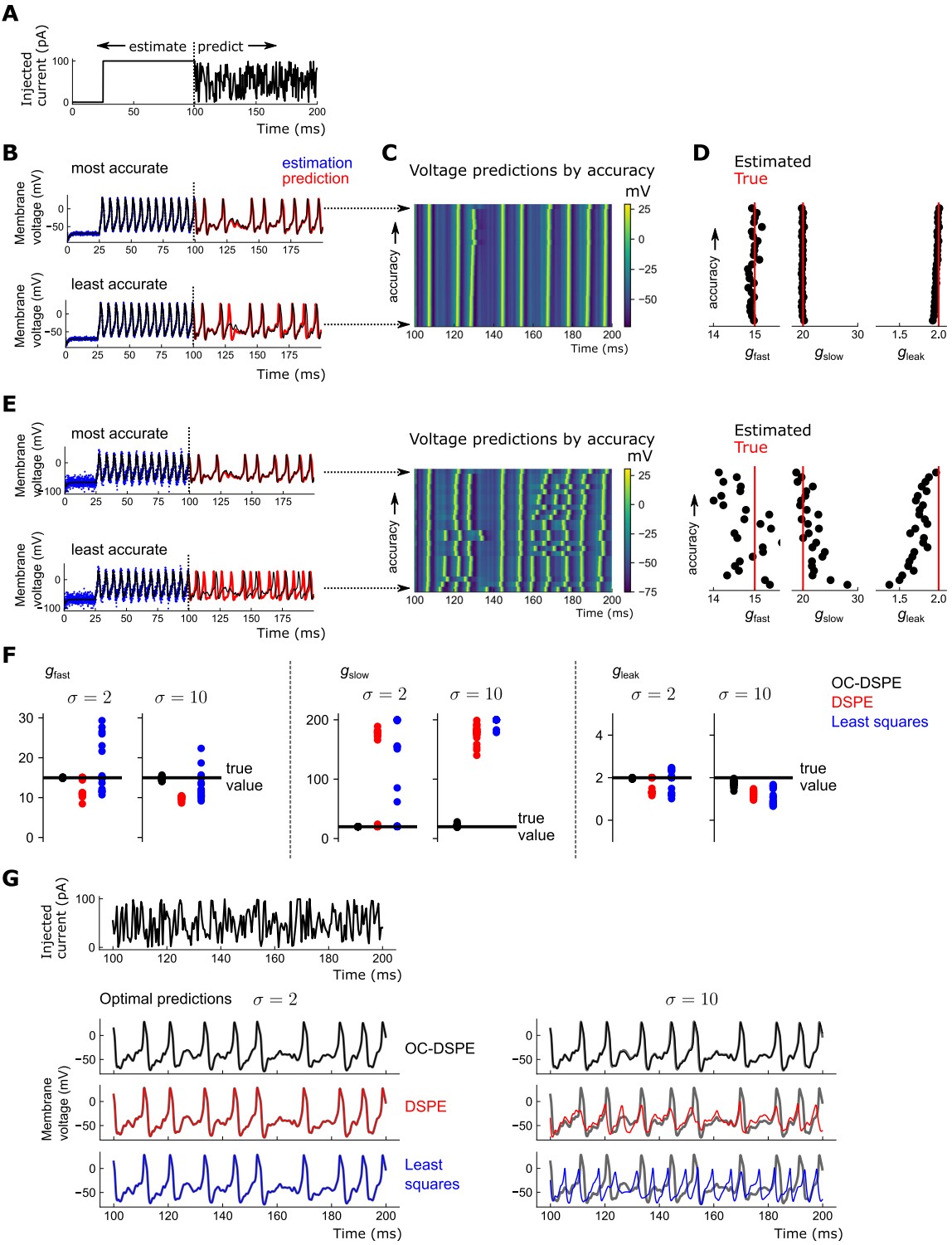

**Fig 4. Estimation of states and linear conductances in Morris-Lecar model.** A: Stimulating current used to estimate parameters and compare forward predictions. B: Most accurate (top) and least accurate (bottom) of 25 estimations using OC-DSPE, $\sigma = 2$ mV. Black: true voltage; blue: noisy observations; red: forward prediction using estimated parameters. C: Heatmap of voltage over time for all 25 runs, with the run giving lowest (highest) prediction error on top (bottom). D: Estimated conductances corresponding to state estimations in C. E: Analogue of B-D, for higher measurement noise $\sigma = 10$ mV. F: Estimations of the linear conductances among the $Q = 25$ runs using OC-DSPE (black), DSPE (red) and constrained least squares (blue), for 2 different levels of measurement noise ($\sigma = 2$, 10 mV, respectively). G: Top trace: Stimulus used for prediction (note timescale is from 100 to 200 ms) for the 3 methods in F. Bottom

left 3 traces: predictions using optimally estimated parameters among the $Q = 25$ runs in F, using the 3 different estimation methods, for $\sigma = 2$ mV measurement noise. Black, red, blue: OC-DSPE, DSPE, and least squares predictions, respectively. Grey: Predicted trajectory using the true parameters (note this is visually indistinguishable from the accurate predictions). Bottom right 3 traces: Same, for $\sigma = 10$ mV measurement noise.

along the spiking limit cycle. Instead, valuable information toward the parameter estimates could be gained by pushing the system to less-visited regions of phase space. One option is to use a staircase signal consisting of many step currents [31]. Alternatively, by using a *dynamic* current, we could persistently modulate the system between different limit cycle manifolds.

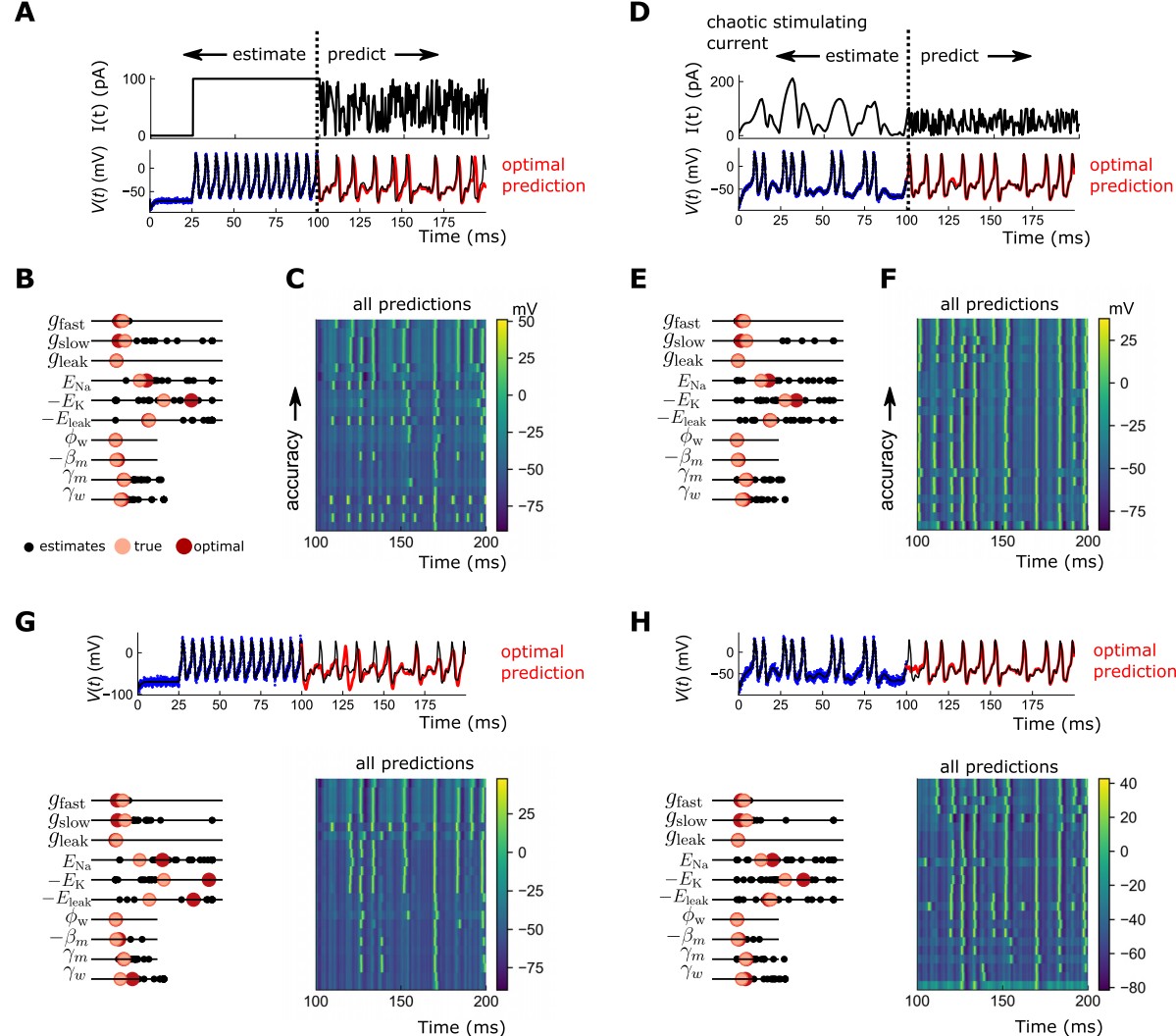

**Fig 5. Estimation of linear and nonlinear parameters in Morris-Lecar model.** A: (top) Stimulating current used for estimation and prediction. (bottom) Black: true voltage trace; blue: noisy observations; red: predicted voltage trace with highest accuracy. B: Estimated parameters for all 25 runs. Black: individual 25 estimates of each parameter; pink: true value of parameter; red: parameter estimate corresponding to lowest prediction error. C: Predicted voltage traces from parameters, from highest accuracy (top) to lowest accuracy (bottom), for all 25 runs. D-F; same as A-C but now using a complex stimulating current. Prediction accuracy is markedly more reliable. G-H: Same as A-F, for higher noise ($\sigma = 10$ mV), for step stimulating current (G) and complex stimulating current (H). In the presence of higher noise, optimal state and parameter estimates are nonetheless very accurate for complex stimulating currents (H).

Since the shape of these manifolds differs by input current, and since steady state is never achieved, the system would cover a larger swath of the phase space through transient motion [18]. This richer sampling of the phase space would increasingly enhance our estimate of $g_i$.

We therefore repeated the experiments using a stimulating current proportional to the $x$ component of the 10D Lorenz96 system (we took the absolute value so all currents were positive, and linearly scaled the time axis) (Fig 5D). In the optimally predicted $\hat{V}(t)$, the shifts in spike times are now absent, and the dropped spike near $t \sim 200$ is recovered (Fig 5D). The estimated parameters are also more precise, producing markedly more robust predictions (Fig 5E and 5F). Finally, we repeat the experiments for the higher measurement noise $\sigma = 10$ mV. The optimal prediction for the step current are of moderate accuracy: many spikes are reconstructed but many are dropped (Fig 5G). For the chaotic driving current, spike times in the optimal prediction match very well against the data (Fig 5H).

Up to this point, we have considered uncorrelated Gaussian noise added *post hoc* to the true membrane voltage. This noise would reflect fluctuations in the recording electrode, but would not reflect other sources of noise inherent to the neuron itself. Alternatively, we can add noise directly into the model equations when generating the data. Such Langevin, or process, noise would reflect noisy current inputs into the neuron, and could produce qualitative changes in neuron response, such as mistimed or even missing action potentials. What is the effect of these major changes on the estimation quality of OC-DSPE? To investigate this, we added a Gaussian-distributed random current input $I_{noise}$ into the Morris-Lecar neuron, and generated observations by integrating this stochastic system forward, and adding measurement noise as before. We chose the statistics of $I_{noise}$ to have mean zero and standard deviation SD = 10, 50, or 100 pA—note that this is up to the magnitude of the injected current (Fig 6A). Indeed, for appreciable noise, SD > 50 pA, the neuron membrane voltage is modulated considerably—spikes are mistimed or even missed (Fig 6A). We then estimated the parameters in the deterministic Morris-Lecar model using OC-DSPE. Optimal predictions over $Q = 25$ runs show that for SD up to 50 pA, the forward predictions (Fig 6B) and parameters (Fig 6C) are highly accurate. For SD = 100 pA, spike timing is mostly preserved, but resting voltages $E_K$ and $E_{Na}$ have larger errors, and some action waveforms are affected. Together, this indicates that even with substantial corrupting model noise, OC-DSPE can accurately infer latent states and model parameters, though there is a breakdown when the fluctuations approach the size of the injected current.

Finally, we investigated the effect of an underdetermined model description, which would reflect, say, incomplete knowledge of which ion channels are present in the neuron. As a simple test, we generated data using the Morris-Lecar model from above, but with an extra ion channel—the *persistent Na channel* $I_{NaP}$ [52]—in addition to the already present K and Na channels. The presence of this channel had measurable effects on neuron response, including missed spikes (Fig 7A). We then estimated model parameters of the original Morris-Lecar model, which had only the Na and K channels (K+Na model), using observations generated from the model with the extra NaP channel (K+Na+NaP model). Despite the presence of an ion channel not accounted for, optimal predictions were highly accurate (Fig 7B). Interestingly, the optimally predicted parameters using K+Na+NaP observations were only slightly perturbed from those fit to observations of the K+Na model (i.e. fits from Fig 5D–5F) (Fig 7C). This indicated that small perturbations in parameter space can appropriately adjust the fit to observations from a different description. The degree to which this is possible in general, is of course, highly dependent on *how* misspecified the model is, but it is comforting that OC-DSPE is robust to a degree of incomplete knowledge in the model description.

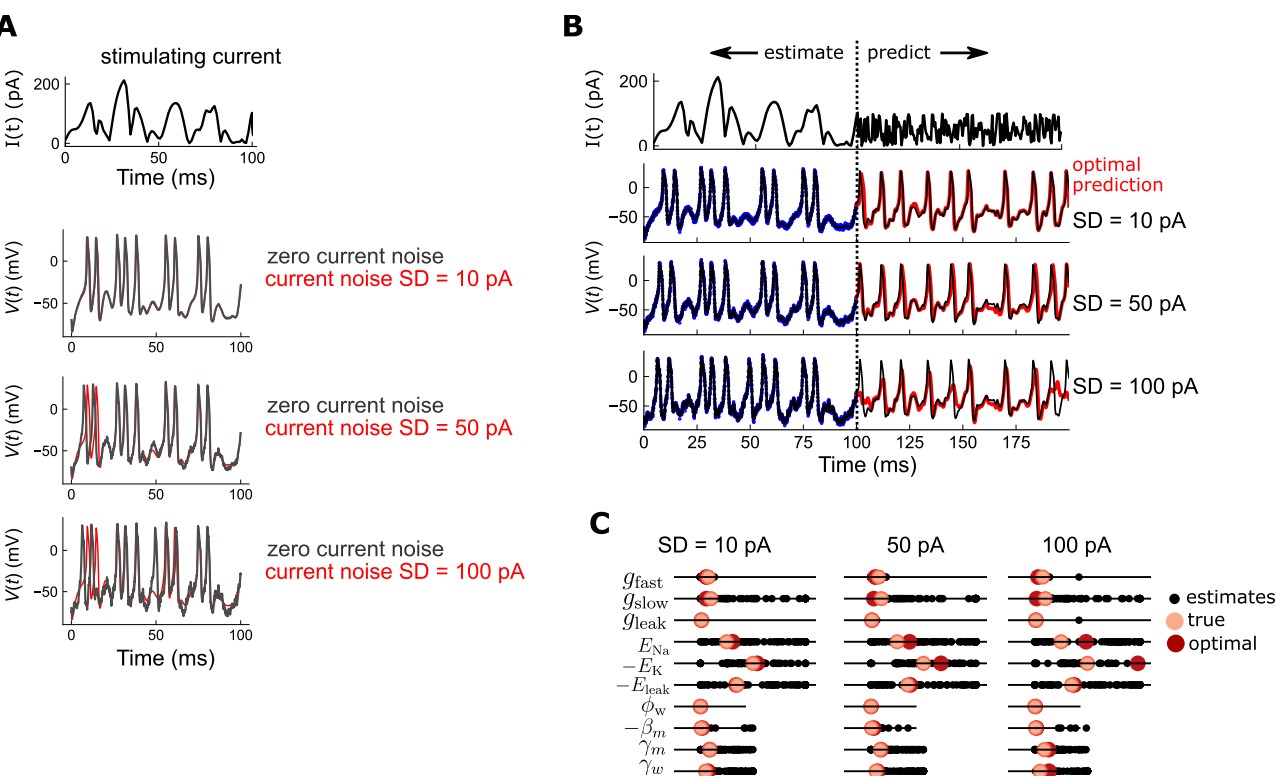

**Fig 6. OC-DSPE estimation in the presence of noisy current inputs.** A: (top trace) Stimulating current used for estimation of parameters. (bottom 3 plots) Resulting voltage traces for the stimulating current, for injected current noise levels of $\sigma$ = 10, 50, 100 pA (light red traces), overlaid with the trace for zero injected current (light black traces). Here, injected current noise acts as a type of model/process noise, since it affects the evolution of the dynamics directly. This contrasts with the "measurement noise" used above, which is uncorrelated, additive noise added to the hidden state space output. B: (top trace) Injected current used for estimation (0 to 100 ms) and prediction (100 to 200 ms). (bottom 3 traces) True trajectory (black), observed data (blue), and optimal estimate (red), for 3 noise levels, respectively, 10, 50, 100 pA. Observed data is generated by integrating the model dynamics with process noise at the level indicated (10, 50, 100 pA), and then adding measurement noise $\sigma$ = 2 mV to the output. C: Predicted parameters for 100 estimations, for 3 model noise levels. Black dots: all estimates; red: parameter corresponding to the optimal estimate; pink: the true parameter estimate.

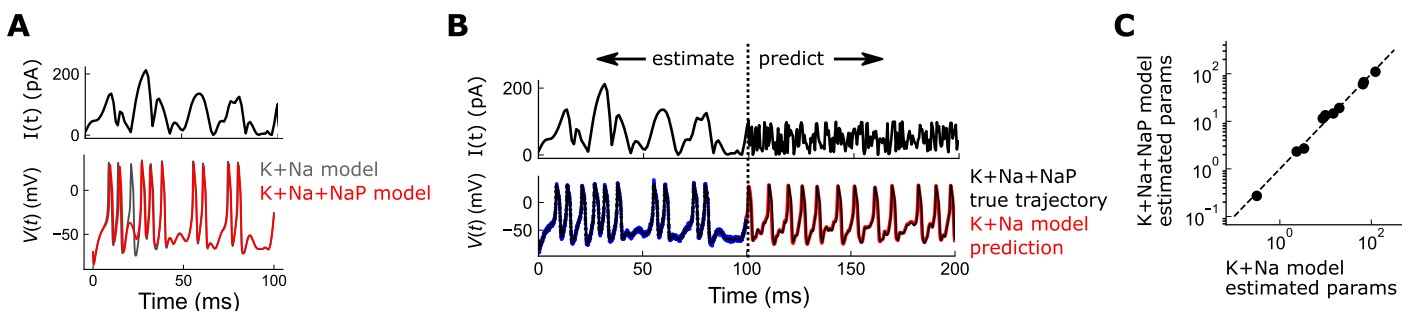

**Fig 7. OC-DSPE performance with an under-specified neuron model.** A: Injected current (top) and resulting membrane voltage (bottom), using the original Morris-Lecar model with Na and K channels (grey), or with an extra NaP channel (red). The addition of the extra channel has appreciable effects on the response, including missing spikes. B: Injected current (top) and optimal OC-DSPE estimated voltage trace (bottom; black) by fitting observations (bottom; blue) from a K+Na+NaP model to an Na+K model. Red: forward predictions of the K+Na model using estimated parameters, overlaid with forward predictions of the true K+Na+NaP model (black). C: The optimal parameters used in B are plotted against the optimal parameters produced by fitting observations without model misspecification (i.e. parameters from Fig 5D–5F). The estimated parameters are nearly in agreement, but small deviations in the optimal fits account for the modified traces that fit the K+Na+NaP observations.

## Discussion

Mathematical models for biological neuron and circuit models are informed intimately by neurobiology. Voltage dynamics in the Hodgkin-Huxley model, for example, obey classical electrodynamic equations for variable capacitors, reflecting the assumption of cell membranes as insulating barriers separating charged species. But while biological considerations may strongly constrain the model description, the many parameters in these models are unspecified. In fact, it is precisely the variability of these parameters that accounts for the vast diversity in system response, and ultimately dictates how these neural systems process external stimuli.

Our approach is to cast the inference problem into an optimal control framework, resulting in a system of coupled ODEs for the dynamics describing the evolution of the optimal estimate. These ODEs, Eq 10, pose two difficulties: they are persistently unstable since they comprise a Hamiltonian system, and they are not directly integrable anyways, since the parameters are not specified. We address these two issues simultaneously by imposing these dynamics as constraints on a related, tractable cost function, and enforcing them iteratively as a penalty term. This is similar to a recent method identifying state space inference with Hamiltonian systems [49], and shares overlaps with a related homotopy approach for parameter estimation in nonlinear systems [42]. These approaches and ours adopt the viewpoint that, rather than i) recursively filtering with linearized dynamics or ii) attempting to represent high-dimensional distributions with particle estimates, the burden of nonlinearity can be shifted to the cost function, provided that nonconvexity can be introduced in a systematic way. Globally optimizing over these variables is, of course, only possible for data that is analyzed offline, as in calcium imaging or electrophysiological measurements.

Several studies in recent years have developed direct optimization-based approaches for state and parameter estimation in neural systems [6, 32–34, 40, 49, 53, 54]. Directly optimizing high-dimensional distributions may seem computationally prohibitive, since the search space is large—equal to the number of dynamical states times the number of measurement times, plus the number of constant parameters. But optimization in state-space models benefits from the highly sparse structure of the Hessian matrix [6], making these methods amenable to fast linear algebraic routines. While our approach does not directly optimize the usual posterior distribution appearing in the Bayesian setting [5, 6], the Hessian matrices required by Newton-type minimizers have a similarly sparse structure, so the computational benefits remain.

The "constrained least squares" estimation we have used for comparison (Figs 3 and 4) is functionally equivalent to the *variational annealing* method used in many recent efforts in state and parameter estimation in chaotic and neural systems [7, 18, 20, 32–34]. Variational annealing performs successive iterations of a cost function that sums model error and measurement error; at each iteration, the weight of the model error term is increased. This method is equivalent to a constrained least squares, since the measurement term is a least squares error, and the model error term is a sum of many constraint equations, each enforcing the model dynamics at each step. By increasing the model error weight iteratively, the constraints are enforced gradually, as in a typical penalty-based constrained optimization. This work illustrates that while variational annealing can be effective for nonlinear parameter inference, it still has significant limitations in the presence of appreciable measurement or model noise—a limitation that is handled far more effectively by OC-DSPE.

An intriguing recent work [31] has explored how suboptimal parameter estimates can be "nudged" out of local minima by adding and then removing artificial measurement noise. That technique produced highly accurate estimations of 41 parameters in a conductance-based neuron model, though compared to our results here, the measurement noise was 10

**Table 1. Comparison of various filtering (sequential) and optimization-based approaches for state and parameter estimation in dynamical systems.**

| Method | Type | States | Params | Notes |
|---|---|---|---|---|
| Kalman filter [5] | sequential | linear | N/A | standard method for linear-Gaussian models |
| Extended/Unscented Kalman filter [5] | sequential | nonlinear | N/A | Allows non-Gaussian noise or nonlinear models |
| Sequential Monte Carlo (SMC) [24] | sequential | nonlinear | N/A | Scales poorly with model dimension |
| Nudging [45] | sequential | nonlinear | N/A | Computationally trivial—only integrate forward model dynamics |
| SMC + Expectation maximization [26] | sequential + linear optimization | nonlinear | linear | Effective for linear parameters and high noise |
| Quasi-static variational assimilation [51] | nonlinear optimization | nonlinear | N/A | Reduces effect of local minima for long time traces |
| DSPE [40] | nonlinear optimization | nonlinear | nonlinear | Estimate many nonlinear parameters, given low measurement noise [19, 31] |
| Variational annealing [33] | nonlinear optimization | nonlinear | nonlinear | Estimate many nonlinear parameters, given appreciable measurement noise [18, 20] |
| OC-DSPE | nonlinear optimization | nonlinear | nonlinear | Effective for many nonlinear parameters with high model or measurement noise; computationally expensive |

Comparison of various methods for state and parameter estimation in dynamical systems, including neural models. The 3rd and 4th column refer to whether the method is applicable to estimated nonlinear states and parameters, respectively. Sequential methods refer to those that iteratively update an estimate, while optimization methods rely on minimizing a cost function. Sequential methods in general tend to be computationally faster, since they require just one or a few iterative passes through the data, scaling linearly with the length of the data. However, they are not directly applicable to estimating parameters. Optimization-based techniques are directly applicable to parameter estimation, and can be aided computationally by sparse Hessians and automatic differentiation. DSPE, variational annealing (akin to constrained least squares), and OC-DSPE can simultaneously estimate nonlinear parameters and dynamical states. OC-DSPE is more accurate, but also more computationally expensive due to the presence of multiple nonlinear constraints.

times smaller. Still, one could envision incorporating such "noise-regulation" in our method here to suppress the impact of local minima.

Our emphasis is on estimation accuracy, rather than computational cost. Indeed, both doubling the state space to a Hamiltonian system and imposing constraints increases computation time compared to, for example, the variational annealing method [32, 33], or directly optimizing the least squares error subject to hard model constraints. Extending this framework to larger systems would require a careful consideration of the tradeoffs between accuracy and speed. A comparison of some recent approaches for estimation in dynamical models, with emphasis on neural systems, is shown in Table 1. In general, parameter estimation is best done in the context of an optimization framework, such as DSPE or variational annealing. Among these, OC-DSPE fares well in terms of accuracy especially with high levels of noise, but with higher computational cost. Still, the accurate estimation of a large number of nonlinear parameters suggests that our method, combined with improved optimization routines, may be relevant for larger neural systems.

For completeness, we note that many methods similar to OC-DSPE, including most listed in Table 1, are often referred to as *data assimilation* methods [55, 56]. Data assimilation is a broad, and unfortunately somewhat nebulous, term generally referring to any technique combining dynamical models with observations. Within this definition, the Kalman Filter or any of its variants is the most well-known. If viewed in this context, OC-DSPE and variational annealing fall into the class of *variational* data assimilation methods, which cast the estimation problem as an optimization problem. Neural data inference are amenable to variational approaches since the observation windows are relatively short and estimation is performed offline—in contrast to weather prediction, where observations come in continuously, the observation windows are long, and predictions must be made in real-time.

There are a number of limitations of, and potential extensions to, OC-DSPE. Throughout, we have treated the measurement matrix as diagonal, where each observation corresponds to a

unique latent state, and noise variances are treated as a fixed and known quantity. This is in line with experimental conditions of intracellular recordings *in vitro*, where the measurement uncertainty can be quantified by fluctuations in the voltage signal when the injected current is zero. Still, this may not be the case for *in vivo* data, where measurements reflect superpositions of multiple neurons in each recording channel, and **H** is unknown and not diagonal. In principle, OC-DSPE could be extended to include the observation matrix **H** as a set of additional inferred parameters, a potential direction for future study.

We have focused here on estimation of parameters in single neuron models, the obvious extension being to the inference of biophysical neuron parameters and connection weights in neural networks. Inference in neural networks will demand a different modeling approach, since highly-resolved voltage waveforms from multiple neurons in vivo are inaccessible, given current technologies. It is unlikely that detailed conductance-based models would be resolvable, and one might opt instead for reduced phenomenological models [1, 26] sharing the desired dynamical features, but containing richer dynamics than simple perceptrons. Our methodology is most relevant to data taken *in vitro*, where currents can be externally controlled. Fruitful directions for future research would carefully consider these limitations, deciding how best to adapt this and other model-based procedures for brain-wide calcium imaging data, as has been explored in recent studies [28].

## Materials and methods

### DSPE cost function generation

To generate the data used in Fig 2A, we numerically integrated the Lorenz96 system in 10 dimensions using the scipy.odeint routine in Python, which runs the FORTRAN solver LSODE. The integration was performed over 501 timepoints from $t_0 = 0$ to $t_N = 8$ at a step of $dt = 0.016$. Observations $y_i$ were generated by adding uncorrelated Gaussian noise $\sim \mathcal{N}(0, 1)$ to the 4 observed states $x_1$, $x_4$, $x_7$, $x_{10}$. The states used in Fig 2B were generated similarly, but using the controlled dynamics Eq 3 with $U_{11} = U_{44} = U_{77} = U_{10,10} = u$.

For the cost surface representation in Fig 2C, we used measurement noise $\sim \mathcal{N}(0, 0.1)$; the observed states were as above. The full cost surface is technically 5-dimensional—the 4 $U_{ll}$ plus the forcing $F$. To plot a visualizable cross section of this surface, we set all $U_{ll}$ to be the same— this is equivalent to projecting the surface along that line.

### Neuron model

The Morris-Lecar ODE dynamics (K+Na model) are given by:

$$
\begin{aligned}
C\dot{V} &= -g_{\text{fast}}m_\infty(V)(V - E_{Na}) - g_{\text{slow}}w(V - E_K) - g_{\text{leak}}(V - E_{\text{leak}}) + I_{\text{stim}} \\
\dot{w} &= \phi_w \frac{w_\infty(V) - w}{\tau_w(V)}
\end{aligned}
\tag{12}
$$

where

$$
\begin{aligned}
w_\infty(V) &= 0.5(1 + \tanh(\frac{V - \beta_{\text{w}}}{\gamma_{\text{w}}})) \\
m_\infty(V) &= 0.5(1 + \tanh(\frac{V - \beta_{\text{m}}}{\gamma_{\text{m}}})) \\
\tau_w(V) &= (\cosh\frac{V - \beta_{\text{w}}}{2\gamma_{\text{w}}})^{-1}
\end{aligned}
\tag{13}
$$

The ground truth parameters were set at as $C = 2.5$, $g_{\text{fast}} = 20$, $g_{\text{slow}} = 15$, $g_{\text{leak}} = 2$, $E_{\text{Na}} = 50$, $E_{\text{K}} = -100$, $E_{\text{leak}} = -70$, $\phi_w = 0.12$, $\beta_w = 0$, $\beta_m = -1.2$, $\gamma_m = 18$, $\gamma_w = 10$. Data used for state and parameter estimation was generated by integrating these dynamics with a prescribed stimulating current $I_{\text{stim}}$, over a window of [0, 100] ms at a timestep of 0.05 ms. For the step stimulus (Figs 4 and 5A–5C and 5G), the current was set to 100 pA. For the chaotic stimulating current (Fig 5D–F and 5H)), we used the absolute value of the output of 1 variable of the 10D Lorenz96 system, scaled in time by a factor of 15 and in value by a factor of 20.

For the Morris Lecar system with persistent Na channel (NaP channel), the dynamics are defined as:

$$
\begin{aligned}
C\dot{V} &= -g_{\text{fast}} m_\infty(V)(V - E_{\text{Na}}) - g_{\text{slow}} w(V - E_{\text{K}}) \\
&\quad - g_{\text{NaP}} m_\infty(V)(V - E_{\text{NaP}}) - g_{\text{leak}}(V - E_{\text{leak}}) + I_{\text{stim}}
\end{aligned}
\tag{14}
$$

$$
\dot{w} = \phi_w \frac{w_\infty(V) - w}{\tau_w(V)}
\tag{15}
$$

where $g_{\text{NaP}} = 3$ and $E_{\text{NaP}} = 50$.

Finally, for the neuron with input current noise, we define the model as:

$$
\begin{aligned}
C\dot{V} &= -g_{\text{fast}} m_\infty(V)(V - E_{\text{Na}}) - g_{\text{slow}} w(V - E_{\text{K}}) \\
&\quad - g_{\text{leak}}(V - E_{\text{leak}}) + I_{\text{stim}} + I_{\text{noice}}
\end{aligned}
\tag{16}
$$

$$
\dot{w} = \phi_w \frac{w_\infty(V) - w}{\tau_w(V)}
\tag{17}
$$

where $I_{\text{noise}}$ at each timestep is chosen from a normal distribution with zero mean and standard deviation SD, where we investigate SD = 10, 50, or 100.

## Constrained optimization

The estimated states and parameters were all found with constrained optimization using the limited-memory Broyden-Fletcher-Goldfarb-Shanno algorithm with constraints (L-BFGS-B), implemented in Python with the package scipy.optimize.minimize. These optimizations are high-dimensional. The search space in DSPE is $ND + NL + P$-dimensional, where $N$ is the number of timepoints, $D$ is the dimension of the dynamical system, $L$ is the number of measurements, and $P$ is the number of parameters. The $NL$ term accounts for the control variables $U_{ll}(t_n)$, one for each observed variable at each timepoint. For least squares, the search space is $ND + P$ dimensional, and for OC-DSPE, the search space is $2ND + P$-dimensional to account for the momenta variables. Bounds must be supplied on all state and parameter variables; for the Lorenz96 system, the states were bounded in [-15, 15], $F$ was bounded in [1, 20], and $U_{ll}(t_n)$ were bounded in [0, 100]. For the Morris Lecar system, $V$ was bounded in [-100, 100], the gating variable $w$ in [0, 1], and the in [0.01, 200]. For OC-DSPE, the momenta $p$ must also be bounded; these were set at [-100, 100].

Equality constraints were required for all 3 methods—least squares, DSPE, and OC-DSPE. These constraints enforced either the raw, uncontrolled system dynamics (least squares; Eq 1), the controlled system dynamics (DSPE; Eq 7), or the estimation dynamics (OC-DSPE; Eq 10). The constraints were enforced iteratively with a penalty method. Specifically, for all 3 routines,

the following function was minimized.

$$C(\beta) = C(\cdot) + 2^\beta \sum_{i,n} R_i g_{i,n}^2(\cdot)$$

(18)

where $C(\cdot)$ is the cost function and $g_{i,n}(\cdot)$ is a discretization of the $i$th constraint equation at timepoint $t_n$, and $\beta$ was iteratively stepped up from 0 to 24. Specifically, $C(\beta = 0)$ was minimized with a randomly chosen initialization of all states and parameters within their bounds; the result of this was used as the initial guess for the minimization of $C(\beta = 1)$, etc. The estimate at $\beta = 24$ was used taken to be the optimal estimate. $C(\cdot)$ for DSPE was Eq 6, for least squares was Eq 6 absent the **U** terms, and for OC-DSPE was Eq 11. The constraints were discretized using Hermite-Simpson quadrature, which results in $N$ constraint equations for each $i$. $R_i$ is a constant factor that allows the individual constraints to scale respectively with the dynamic range of that particular variable; this normalizes the contributions of different variables to the penalty term. For the Lorenz system $R_i = 1e - 4, \forall i$; for the neuron model, $R_V = 1e - 4, R_w = 1, R_{p_V} = 1, R_{p_w} = 1$. Note that the least squares method with iteratively enforced constraints is equivalent to the *variational annealing* algorithm proposed previously [18, 32–34, 49, 54]. These estimations were done many times in parallel (100 for Lorenz96 system; 25 for the neuron model) with different initial guesses for the parameters and states at $\beta = 0$.

For the Morris-Lecar neuron model, once the optimal state and parameter estimates are determined, they were used to generate predictions in the interval [100, 200] by integrating forward the dynamics with the final state estimate $\mathbf{x}(t_N)$ using the estimated parameter values.

All optimization routines required first derivatives of the cost function and the constraints; we obtained these in a few lines of code by using the automatic differentiation package PyAdolc.

## OC-DSPE estimation dynamics

The optimal control conditions were derived using the Pontryagin Minimum Principle [43]. In this formulation, one must prescribe the cost function along with the system dynamics; each of these may depend on the states, parameters, and controls. Our cost function is the DPE cost function Eq 6 (in continuous time)—a least squares matching of observations and states, plus a quadratic control penalty (restatement of Eq 8 from the main text):

$$C = \int_0^T dt \mathcal{L}(\mathbf{x}, \mathbf{U}) \quad = \int_0^T dt \left[\frac{1}{2}||\mathbf{Hx} - \mathbf{y}||^2 + ||\mathbf{U}||^2\right],$$

where $\mathcal{L}$ is the system "Lagrangian." The control optima are derived from the corresponding "Hamiltonian" function which is defined from the Lagrangian via:

$$\mathcal{H}(\mathbf{x}, \mathbf{U}, \mathbf{p}) = \mathcal{L}(\mathbf{x}, \mathbf{U}) + \mathbf{p} \cdot \mathbf{f}(\mathbf{x}, \mathbf{U}, \mathbf{\Theta}),$$

(19)

where $\mathbf{p}$ is conjugate momentum variable. Note that $\mathbf{x}$, $\mathbf{U}$, and $\mathbf{p}$ are all time-dependent, with the time-dependence suppressed for readability. The Pontryagin Minimum Principle states that then the optimal control $\mathbf{U}_{\text{opt}}$ along the integral curves of this system must satisfy:

$$\mathbf{U}_{\text{opt}} = \arg \min_{\mathbf{U}} \mathcal{H}(\mathbf{x}, \mathbf{U}, \mathbf{p}).$$

(20)

If the optimal control exists in the interior of the space of admissable controls, then the

minimum of Eq 20 can be found from the stationary points of $\mathcal{H}$ with respect to $\mathbf{U}$:

$$0 \quad = \frac{\partial \mathcal{H}}{\partial \mathbf{U}}, \tag{21}$$

which, for the nonzero diagonal elements of $\mathbf{U}$ satisfy

$$0 = \frac{\partial H}{\partial U_{ll}} = U_{ll} + p_l [\mathbf{y} - \mathbf{H}\mathbf{x}]_l \tag{22}$$

In addition to the condition on the stationarity of $\mathcal{H}$, the system dynamics must satisfy Hamilton's equations, one of which reproduces the state space dynamics $\dot{\mathbf{x}}$, while the second gives the dynamics for the conjugate momenta (writing out the indices explicitly):

$$\dot{x}_d \qquad = \frac{\partial \mathcal{H}}{\partial p_d} = f_{\mathrm{cont},d}(\mathbf{x}, \mathbf{U}, \mathbf{\Theta})$$
$$\dot{p}_d \quad = -\frac{\partial \mathcal{H}}{\partial x_d} = -\frac{\partial \mathcal{L}}{\partial x_d} - \frac{\partial \mathbf{f}_{\mathrm{cont}}(\mathbf{x}, \mathbf{U}, \mathbf{\Theta})}{\partial x_d} \cdot \mathbf{p} \tag{23}$$

Writing out the derivatives of $\mathcal{L}$, and expanding out $f_{\mathrm{cont}}$, this gives the following:

$$\dot{x}_d \qquad = f_d(\mathbf{x}, \mathbf{\Theta}) + [\mathbf{U} \cdot (\mathbf{y} - \mathbf{H}\mathbf{x})]_d$$
$$\dot{p}_d \quad = -[\mathbf{H}^T(\mathbf{H}\mathbf{x} - \mathbf{y})]_d + [\mathbf{H}^T \mathbf{U}^T \mathbf{p}]_d - \frac{\partial \mathbf{f}(\mathbf{x}, \mathbf{U}, \mathbf{\Theta})}{\partial x_d} \cdot \mathbf{p} \tag{24}$$

The momenta equations $\dot{\mathbf{p}}$ provide dynamical constraints on the control variables $U_{aa}$, absent from nudging and DSPE. For observable states, the optimality condition Eq 22 can be inverted for the momenta $p_d$ (Eq 9). Applying this in Eq 24 gives the complete set of optimally-controlled dynamics, in Hamiltonian coordinates $\{\mathbf{x}, \mathbf{p}\}$, Eq 10.

## Acknowledgments

We thank Paul Rozdeba for helpful discussions, and Viraaj Jayaram and Gustavo Madeira Santana for a careful reading of the manuscript.

## Author Contributions

**Conceptualization:** Nirag Kadakia.

**Data curation:** Nirag Kadakia.

**Formal analysis:** Nirag Kadakia.

**Funding acquisition:** Nirag Kadakia.

**Investigation:** Nirag Kadakia.

**Methodology:** Nirag Kadakia.

**Resources:** Nirag Kadakia.

**Software:** Nirag Kadakia.

**Validation:** Nirag Kadakia.

**Writing – original draft:** Nirag Kadakia.

**Writing – review & editing:** Nirag Kadakia.

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
