## [Decision Letter · Decision Letter 0]

30 Mar 2022

Dear Dr. Kadakia,

Thank you very much for submitting your manuscript "Optimal control methods for nonlinear parameter estimation in biophysical neuron models" for consideration at PLOS Computational Biology.

As with all papers reviewed by the journal, your manuscript was reviewed by members of the editorial board and by several independent reviewers. In light of the reviews (below this email), we would like to invite the resubmission of a significantly-revised version that takes into account the reviewers' comments.

All three reviewers have recommended substantial revisions to be made to the manuscript. The manuscript is generally well-written and timely, however the methodology needs clarifications and the work needs to be better situated in its background.

We cannot make any decision about publication until we have seen the revised manuscript and your response to the reviewers' comments. Your revised manuscript is also likely to be sent to reviewers for further evaluation.

Sincerely,

Alain Nogaret, PhD

Guest Editor

PLOS Computational Biology

Lyle Graham

Deputy Editor

PLOS Computational Biology

All three reviewers have recommended substantial revisions to be made to the manuscript. The manuscript is generally well-written and timely, however the methodology needs clarifications and the work needs to be better situated in its background.

Reviewer's Responses to Questions

**Comments to the Authors:**

Reviewer #1: N/A

Reviewer #2: This paper introduces a new method for inferring the state and parameters of nonlinear stochastic dynamical systems where there are fewer measurements than the number of states. The method, termed OC-DSPE, combines a dynamic nudging approach with optimal control. By applying OC-DSPE to the chaotic Lorenz96 system and the Morris-Lecar neuron model, the author shows that the new method is superior to ordinary least squares and DSPE without the optimal-control component.

The paper is well written. The approach and result description, including figures, is clear. The method seems to offer a useful new approach to an important general problem in neuroscience. There are however several major issues regarding the method’s applicability and advantages in the context of neural data analysis that are not adequately addressed. Additional analyses based on the points raised below are therefore crucial for appreciating the usefulness of the new approach in answering neuroscientific questions:

1. Noise: the paper’s objective as stated in the abstract is joint state and parameter estimation in neural State Space Models (SSM). These are usually understood to refer to doubly stochastic dynamical systems. However, only measurement noise is considered, while neural recordings also often include process noise as described by the Langevin equation (Eq. 2). How does OC-DSPE fair in such scenario? Besides, only isotropic Gaussian measurement noise is considered, and its variance is assumed to be known. Can noise variance be also inferred as a model parameter? How about nondiagonal covariance matrices?

2. Measurement: Each measurement here is assumed to equal one state plus noise. However, in in vivo recordings from, e.g., multielectrode arrays, measurements usually come from multi-unit activity. They are therefore (an approximately linear) mixture of several states. Can OC-DSPE handle such a situation and infer a mixing matrix as an unknown parameter, i.e., the output matrix in SSMs?

3. Dynamics: The method assumes a full knowledge of the state dynamics. However, in neural recordings not all conductance types present in each neuron type are known a priori. What is OC-DSPE’s solution to underdefined state equations?

4. Predictions: One major advantage of SSMs is their ability to conduct in silico experiments and make predictions of future dynamics (for instance, how a neuron would behave under stimulation conditions that were not used during the inference stage). Are there advantages for using OC-DSPE in such analyses? That the method is more accurate suggests that it may make better predictions, but it would be useful to see that in practice.

5. Comparisons: OC-DSPE was compared to simpler methods and was shown to perform better. However, the paper mentions other approaches that are similar in methodology (quasi-static variational assimilation and variational annealing). What are the advantages of OC-DSPE over this class of methods?

Reviewer #3: This paper proposes a new method, based on chaotic synchronization and optimal control theory, for inferring the parameters of conductance-based neuronal models directly from voltage traces. Results are presented demonstrating that the proposed method, OC-DSPE, yielded more accurate parameter estimates from synthetic data generated by the Lorenz96 system (when only 2 of the 10 state variables are observed) compared to 2 prior techniques. The OC-DSPE method is then used to estimate linear and nonlinear parameters from synthetic voltage traces generated by the Morris-Lecar neuronal model in response to a current pulse and a chaotic stimulating current.

Comments:

This paper is well-written and addresses an important challenge in computational neuroscience. While conductance-based models can be fit to data from voltage-clamp recordings on individual ionic currents, it is often the case that the only data available for model fitting are voltage traces from current-clamp recordings. This is a very difficult parameter estimation problem since only one of the model’s many state variables is directly observed. Thus, an improved method for addressing this challenge would be a significant contribution to the field. However, some major changes to the manuscript are needed to demonstrate that the proposed method does indeed address this challenge.

1) The manuscript includes a direct comparison of the performance of the proposed OC-DPSE method to the prior methods of DSPE and least-squares minimization on data from the Lorenz96 system, but no such comparison is provided for data from the Morris-Lecar model. Since the Lorenz96 system is not a neuronal model, and the focus of the manuscript is on parameter estimation in biophysical neuron models, it would significantly strengthen the manuscript if a direct comparison of the performance for the proposed and prior techniques were provided for the Morris-Lecar model as well.

2) The last paragraph in the “Optimally-controlled dynamical parameter inference” section of the Results (lines 202-220) does a nice job of relating the OC-DSPE method to previous approaches utilizing homotopy continuation and extensions of variational annealing to Hamiltonian manifolds. It ends by saying that Algorithm 1 is in much the same spirit as [49]. It would be helpful to add a concluding sentence here indicating what makes the proposed OC-DSPE method distinct from these previous approaches.

3) The manuscript emphasizes that the proposed method is tailored to situations with considerable measurement noise, however the type of noise used in the “Numerical experiments with the Morris-Lecar model”, where uncorrelated Gaussian noise is added to the ground truth voltage, does not seem very biological to me as it doesn’t appear to affect the timing of action potentials. Is this type of noise present in in vitro current-clamp electrophysiology recordings? A more relevant type of noise, it seems to me, would be some form of dynamical noise (representing unmodeled synaptic inputs, or ion channel noise perhaps) that causes irregularities in the timing of action potentials. Would this type of noise still be considered measurement noise or instead some kind of model noise, and would OC-DSPE still perform well in the face of it? Even if results are not shown with this type of noise, a discussion of the characteristics of the noise encountered in actual experimental voltage traces and how they relate to the numerical experiments shown here and the way that noise is incorporated into the OC-DSPE framework is warranted.

4) Does the proposed OC-DSPE method fall under the general umbrella of data assimilation methods? I’m asking because many of the references cited (11 of them by my count) contain the term data assimilation (or a variant of it) in their title, yet the word assimilation only appears once in the manuscript itself (on line 216). It might help orient readers with regards to the existing literature if it is explicitly stated whether OC-DSPE should be considered as a new variational data assimilation algorithm, or as something else.

5) I have two questions about lines 97-99 which reads “As will be discussed below, the more pressing issue is that sequential estimators such as both nudging and the Kalman filter are not directly suited for the estimation of static parameters.”

First, is it appropriate to refer to nudging as a sequential estimator? I ask because in the Introduction, it seems that chaotic nudging synchronization is included in the paragraph that starts on line 38 as “An alternative to all of these sequential filtering approaches is the direct optimization of a posterior distribution over, jointly, the states at all timepoints and the unknown parameters.”

This also leads to my second question, which is where is it discussed below that nudging and the Kalman filter are not directly suited for the estimation of static parameters? I found this confusing because line 170 says that DSPE is “built naturally on nudging synchronization” and earlier in the Introduction references are cited [5, 25, 26] that use a sequential filtering approach for parameter estimation. Perhaps, rather than alluding to a discussion that will come later, it would be good to include a sentence or two about why nudging and the Kalman filter are not directly suited for the estimation of static parameters immediately following lines 98-99.

Minor comments:

1) Abstract, second sentence: replace “neurons models” with “neuron models”

2) Line 76: replace “an D-dimensional” with “a D-dimensional”

3) Line 170: replace “DSPE built” with “DSPE is built”

4) Equation 9: replace “observed),” with “observed).”

5) Line 185: replace “the the” with “the”

6) Line 250: replace “near on” with “near” or “on”

7) Line 294: extra space after “1” before the period

8) Line 315: replace “as” with “is”

9) Lines 376 and 379: the opening quotation mark is reversed in “nudged” and “noise-regulation”

10) Line 489: missing space between the comma and “and”

**Have the authors made all data and (if applicable) computational code underlying the findings in their manuscript fully available?**

Reviewer #1: **No: **Computational method need further clarification as specified in the review.

Reviewer #2: **No: **The paper mentions a GitHub repository but no link is provided.

Reviewer #3: **No: **In the Data and Code Availability section of the coversheet it says "All codes are available on GitHub" but I didn't see any mention of the specific GitHub repository in the manuscript itself

PLOS authors have the option to publish the peer review history of their article (what does this mean?). If published, this will include your full peer review and any attached files.

Reviewer #1: No

Reviewer #2: **Yes: **Hazem Toutounji

Reviewer #3: No
---

## [Decision Letter · Decision Letter 1]

10 Aug 2022

Dear Dr Kadakia,

We are pleased to inform you that your manuscript 'Optimal control methods for nonlinear parameter estimation in biophysical neuron models' has been provisionally accepted for publication in PLOS Computational Biology.

We also would like you to address the minor comments from the two reviewers.

Best regards,

Alain Nogaret, PhD

Guest Editor

PLOS Computational Biology

Lyle Graham

Deputy Editor

PLOS Computational Biology

Thank you for revising the manuscript and addressing the reviewer comments. I am pleased to recommend publication. The manuscript still contains a small number of typos some of which were listed by the reviewers. These may be corrected the galley proof stage.

Reviewer's Responses to Questions

**Comments to the Authors:**

Reviewer #2: The author address all my comments and made substantial improvement to the article. There remains 3 very minor comments:

Author summary says the approach compares well to other methods but doesn’t mention the advantage and disadvantage of the new method (ie handling noise better but higher computational cost). hugely-dimensional  high dimensional

Line 369: deterministic Morris-Lecar model – isn’t it stochastic?

Line 448: compute time  computation or computing time

Reviewer #3: Author summary, line 2: replace “behavioral response” with “behavioral responses” or “a behavioral response”

Pg 2, line 29: replace “combines” with "combine”

Pg 3, line 59: replace “parameters” with “parameter”

Pg 12, line 322: replace “estimated of all” with “estimated all”

Pg 18, line 441: replace “and the removing” with “and then removing”

Pg 18, line 470: replace “extensions, to OC-DSPE” with “extensions to, OC-DSPE”

**Have the authors made all data and (if applicable) computational code underlying the findings in their manuscript fully available?**

Reviewer #2: Yes

Reviewer #3: Yes

PLOS authors have the option to publish the peer review history of their article (what does this mean?). If published, this will include your full peer review and any attached files.

Reviewer #2: **Yes: **Hazem Toutounji

Reviewer #3: **Yes: **Casey O. Diekman

---

## [Editor Report · Acceptance letter]

5 Sep 2022

PCOMPBIOL-D-22-00050R1 

Optimal control methods for nonlinear parameter estimation in biophysical neuron models

Dear Dr Kadakia,

I am pleased to inform you that your manuscript has been formally accepted for publication in PLOS Computational Biology. Your manuscript is now with our production department and you will be notified of the publication date in due course.

With kind regards,

Zsofia Freund
